# Provably Robust Metric Learning

**Lu Wang**[1,2]   **Xuanqing Liu**[3]   **Jinfeng Yi**[2]   **Yuan Jiang**[1]   **Cho-Jui Hsieh**[3]
[1]National Key Laboratory for Novel Software Technology,
Nanjing University, Nanjing 210023, China
[2]JD.com, Beijing 100101, China
[3]Department of Computer Science, University of California, Los Angeles, CA 90095
`wangl@lamda.nju.edu.cn` / `wangl@jd.com`   `xqliu@cs.ucla.edu`
`yijinfeng@jd.com`   `jiangy@lamda.nju.edu.cn`   `chohsieh@cs.ucla.edu`

## Abstract

Metric learning is an important family of algorithms for classification and similarity search, but the robustness of learned metrics against small adversarial perturbations is less studied. In this paper, we show that existing metric learning algorithms, which focus on boosting the clean accuracy, can result in metrics that are less robust than the Euclidean distance. To overcome this problem, we propose a novel metric learning algorithm to find a Mahalanobis distance that is robust against adversarial perturbations, and the robustness of the resulting model is certifiable. Experimental results show that the proposed metric learning algorithm improves both certified robust errors and empirical robust errors (errors under adversarial attacks). Furthermore, unlike neural network defenses which usually encounter a trade-off between clean and robust errors, our method does not sacrifice clean errors compared with previous metric learning methods.

## 1   Introduction

Metric learning has been an important family of machine learning algorithms and has achieved successes on several problems, including computer vision [27, 19, 20], text analysis [30], meta learning [44, 40] and others [39, 52, 54]. Given a set of training samples, metric learning aims to learn a good distance measurement such that items in the same class are closer to each other in the learned metric space, which is crucial for classification and similarity search. Since this objective is directly related to the assumption of nearest neighbor classifiers, most of the metric learning algorithms can be naturally and successfully combined with $K$-Nearest Neighbor ($K$-NN) classifiers.

Adversarial robustness of machine learning algorithms has been studied extensively in recent years due to the need of robustness guarantees in real world systems. It has been demonstrated that neural networks can be easily attacked by adversarial perturbations in the input space [43, 18, 2], and such perturbations can be computed efficiently in both white-box [4, 33] and black-box settings [7, 21, 10, 46]. To tackle this issue, many defense algorithms have been proposed to improve the robustness of neural networks [29, 33]. Although these algorithms can successfully defend from standard attacks, it has been shown that many of them are vulnerable under stronger attacks when the attacker knows the defense mechanisms [4]. Therefore, recent research in adversarial defense of neural networks has shifted to the concept of "certified defense", where the defender needs to provide a certification that no adversarial examples exist within a certain input region [50, 12, 55].

In this paper, we consider the problem of learning a metric that is robust against adversarial input perturbations. It has been shown that nearest neighbor classifiers are not as robust as expected [36, 45, 38], where a small and human imperceptible perturbation in the input space can fool a $K$-NN classifier, thus it is natural to investigate how to obtain a metric that improves the adversarial

robustness. Despite being an important and interesting research problem to tackle, to the best of our knowledge it has not been studied in the literature. There are several caveats that make this a hard problem: 1) attack and defense algorithms for neural networks often rely on the smoothness of the corresponding functions, while $K$-NN is a discrete step function where the gradient does not exist. 2) Even evaluating the robustness of $K$-NN with the Euclidean distance is harder than neural networks — attack and verification for $K$-NN are nontrivial and time consuming [45]. Furthermore, none of the existing work have considered general Mahalanobis distances. 3) Existing algorithms for evaluating the robustness of $K$-NN, including attack [53] and verification [45], are often non-differentiable, while training a robust metric will require a differentiable measurement of robustness.

To develop a provably robust metric learning algorithm, we formulate an objective function to learn a Mahalanobis distance, parameterized by a positive semi-definite matrix $\boldsymbol{M}$, that maximizes the minimal adversarial perturbation on each sample. However, computing the minimal adversarial perturbation is intractable for $K$-NN, so to make the problem solvable, we propose an efficient formulation for lower-bounding the minimal adversarial perturbation, and this lower bound can be represented as an explicit function of $\boldsymbol{M}$ to enable the gradient computation. We further develop several tricks to improve the efficiency of the overall procedure. Similar to certified defense algorithms in neural networks, the proposed algorithm can provide a certified robustness improvement on the resulting $K$-NN model with the learned metric. Decision boundaries of 1-NN with different Mahalanobis distances for a toy dataset (with only four orange triangles and three blue squares in a two-dimensional space) are visualized in Figure 1. It can be observed that the proposed Adversarial Robust Metric Learning (ARML) method can obtain a more "robust" metric on this example.

We conduct extensive experiments on six real world datasets and show that the proposed algorithm can improve both certified robust errors and the empirical robust errors (errors under adversarial attacks) over existing metric learning algorithms.

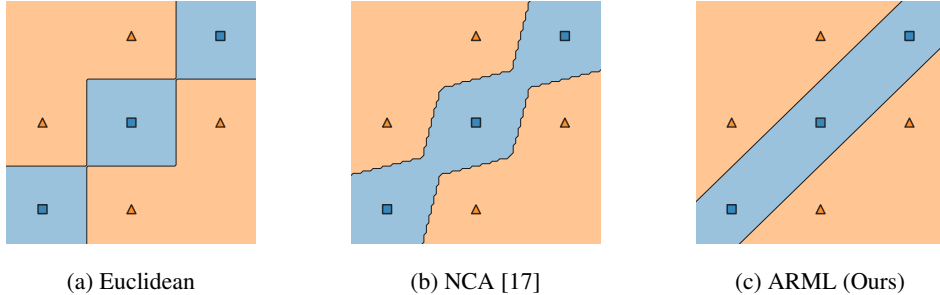

(a) Euclidean        (b) NCA [17]        (c) ARML (Ours)

Figure 1: Decision boundaries of 1-NN with different Mahalanobis distances.

## 2 Background

**Metric learning for nearest neighbor classifiers**    A nearest-neighbor classifier based on a Mahalanobis distance could be characterized by a training dataset and a positive semi-definite matrix. Let $\mathbb{X} = \mathbb{R}^D$ be the instance space, $\mathbb{Y} = [C]$ the label space where $C$ is the number of classes. $\mathbb{S} = \{(\boldsymbol{x}_i, y_i)\}_{i=1}^N$ is the training set with $(\boldsymbol{x}_i, y_i) \in \mathbb{X} \times \mathbb{Y}$ for every $i \in [N]$. $\boldsymbol{M} \in \mathbb{R}^{D \times D}$ is a positive semi-definite matrix. The Mahalanobis distance for any $\boldsymbol{x}, \boldsymbol{x}' \in \mathbb{X}$ is defined as

$$d_{\boldsymbol{M}}(\boldsymbol{x}, \boldsymbol{x}') = (\boldsymbol{x} - \boldsymbol{x}')^\top \boldsymbol{M} (\boldsymbol{x} - \boldsymbol{x}'), \tag{1}$$

and a Mahalanobis $K$-NN classifier $f : \mathbb{X} \to \mathbb{Y}$ will find the $K$ nearest neighbors of the test instance in $\mathbb{S}$ based on the Mahalanobis distance, and then predicts the label based on majority voting of these neighbors.

Many metric learning approaches aim to learn a good Mahalanobis distance $\boldsymbol{M}$ based on training data [17, 13, 48, 22, 42] (see more discussions in Section 5). However, none of these previous methods are trying to find a metric that is robust to small input perturbations.

**Adversarial robustness and minimal adversarial perturbation**    There are two important concepts in adversarial robustness: adversarial attack and adversarial verification (or robustness verification). Adversarial attack aims to find a perturbation to change the prediction, and adversarial

verification aims to find a radius within which no perturbation could change the prediction. Both of them can be reduced to the problem of finding the minimal adversarial perturbation. For a classifier $f$ on an instance $(\boldsymbol{x}, y)$, the *minimal adversarial perturbation* can be defined as

$$\arg\min_{\boldsymbol{\delta}} \|\boldsymbol{\delta}\| \ \text{ s.t. } f(\boldsymbol{x} + \boldsymbol{\delta}) \neq y, \tag{2}$$

which is the smallest perturbation that could lead to "misclassification". Note that if $(\boldsymbol{x}, y)$ is not correctly classified, the minimal adversarial perturbation is $\mathbf{0}$, i.e., the zero vector. Let $\boldsymbol{\delta}^*(\boldsymbol{x}, y)$ denote the optimal solution and $\epsilon^*(\boldsymbol{x}, y) = \|\boldsymbol{\delta}^*(\boldsymbol{x}, y)\|$ the optimal value. Obviously, $\boldsymbol{\delta}^*(\boldsymbol{x}, y)$ is also the solution of the optimal adversarial attack, and $\epsilon^*(\boldsymbol{x}, y)$ is the solution of the optimal adversarial verification. For neural networks, it is often NP-complete to solve (2) exactly [23], so many efficient algorithms have been proposed for attack [18, 4, 3, 10] and verification [50, 49, 35], corresponding to computing upper and lower bounds of the minimal adversarially perturbation respectively. However, these methods do not work for discrete models such as nearest neighbor classifiers.

In this paper our algorithm will be based on a novel derivation of a lower bound of the minimal adversarial perturbation for Mahalanobis $K$-NN classifiers. To the best of our knowledge, there has been no previous work tackling this problem. Since the Mahalanobis $K$-NN classifier is parameterized by a positive semi-definite matrix $\boldsymbol{M}$ and the training set $\mathbb{S}$, we further let the optimal solution $\boldsymbol{\delta}^*_{\mathbb{S}}(\boldsymbol{x}, y; \boldsymbol{M})$ and the optimal value $\epsilon^*_{\mathbb{S}}(\boldsymbol{x}, y; \boldsymbol{M})$ explicitly indicate their dependence on $\boldsymbol{M}$ and $\mathbb{S}$. In this paper we will consider $\ell_2$ norm in (2) for simplicity.

**Certified and empirical robust errors**  Let $\underline{\epsilon}^*(\boldsymbol{x}, y)$ be a lower bound of the norm of the minimal adversarial perturbation $\epsilon^*(\boldsymbol{x}, y)$, possibly computed by a robustness verification algorithm. For a distribution $\mathcal{D}$ over $\mathbb{X} \times \mathbb{Y}$, the **certified robust error** with respect to the given radius $\epsilon \geq 0$ is defined as the probability that $\underline{\epsilon}^*(\boldsymbol{x}, y)$ is not greater than $\epsilon$, namely

$$\mathrm{cre}(\epsilon) = \mathbb{E}_{(\boldsymbol{x}, y) \sim \mathcal{D}}[\mathbf{1}\{\underline{\epsilon}^*(\boldsymbol{x}, y) \leq \epsilon\}]. \tag{3}$$

Note that in the case with $\underline{\epsilon}^*(\boldsymbol{x}, y) = \epsilon^*(\boldsymbol{x}, y)$, the certified robust error at $\epsilon = 0$ is reduced to the *clean error* (the normal classification error). In this paper we will investigate how to compute the certified robust error for Mahalanobis $K$-NN classifiers.

On the other hand, adversarial attack algorithms are trying to find a feasible solution of (2), denoted as $\hat{\boldsymbol{\delta}}(\boldsymbol{x}, y)$, which will give an upper bound, i.e., $\|\hat{\boldsymbol{\delta}}(\boldsymbol{x}, y)\| \geq \epsilon^*(\boldsymbol{x}, y)$. Based on the upper bound, we can measure the **empirical robust error** of a model by

$$\mathrm{ere}(\epsilon) = \mathbb{E}_{(\boldsymbol{x}, y) \sim \mathcal{D}}[\mathbf{1}\{\|\hat{\boldsymbol{\delta}}(\boldsymbol{x}, y)\| \leq \epsilon\}]. \tag{4}$$

Since $\hat{\boldsymbol{\delta}}(\boldsymbol{x}, y)$ is computed by an attack method, the empirical robust error is also called the *attack error* or the *attack success rate*. A family of decision-based attack methods, which view the victim model as a black-box, can be used to attack Mahalanobis $K$-NN classifiers [3, 9, 10].

## 3 Adversarially robust metric learning

The objective of *adversarially robust metric learning* (ARML) is to learn the matrix $\boldsymbol{M}$ via the training data $\mathbb{S}$ such that the resulting Mahalanobis $K$-NN classifier has small certified and empirical robust errors.

### 3.1 Basic formulation

The goal is to learn a positive semi-definite matrix $\boldsymbol{M}$ to minimize the certified robust training error. Since the certified robust error defined in (3) is non-smooth, we replace the indicator function by a loss function. The resulting objective can be formulated as

$$\min_{\boldsymbol{G} \in \mathbb{R}^{D \times D}} \frac{1}{N} \sum_{i=1}^{N} \ell\left(\epsilon^*_{\mathbb{S} - \{(\boldsymbol{x}_i, y_i)\}}(\boldsymbol{x}_i, y_i; \boldsymbol{M})\right) \ \text{ s.t. } \boldsymbol{M} = \boldsymbol{G}^\top \boldsymbol{G}, \tag{5}$$

where $\ell : \mathbb{R} \to \mathbb{R}$ is a monotonically non-increasing function, e.g., the hinge loss $[1 - \epsilon]_+$, exponential loss $\exp(-\epsilon)$, logistic loss $\log(1 + \exp(-\epsilon))$, or "negative" loss $-\epsilon$. We also employ the matrix $\boldsymbol{G}$ to

enforce $M$ to be positive semi-definite, and it is possible to derive a low-rank $M$ by constraining the shape of $G$. Note that the minimal adversarial perturbation is defined on the training set excluding $(\boldsymbol{x}_i, y_i)$ itself, since otherwise a 1-nearest neighbor classifier with any distance measurement will have 100% accuracy. In this way, we minimize the "leave-one-out" certified robust error. The remaining problem is how to exactly compute or approximate $\epsilon_\mathbb{S}^*(\boldsymbol{x}, y; \boldsymbol{M})$ in our training objective.

### 3.2 Bounding minimal adversarial perturbation for Mahalanobis $K$-NN

For convenience, suppose $K$ is an odd number and denote $k = (K+1)/2$. In the binary classification case for simplicity, i.e., $C = 2$, the computation of $\epsilon_\mathbb{S}^*(\boldsymbol{x}_{\text{test}}, y_{\text{test}}; \boldsymbol{M})$ for Mahalanobis $K$-NN could be formulated as

$$
\min_{\substack{\mathbb{J} \subseteq \{j : y_j \neq y_{\text{test}}\}, |\mathbb{J}| = k \\ \mathbb{I} \subseteq \{i : y_i = y_{\text{test}}\}, |\mathbb{I}| = k-1}} \min_{\boldsymbol{\delta}_{\mathbb{I}, \mathbb{J}}} \|\boldsymbol{\delta}_{\mathbb{I}, \mathbb{J}}\|
$$

$$
\text{s.t. } d_{\boldsymbol{M}}(\boldsymbol{x}_{\text{test}} + \boldsymbol{\delta}_{\mathbb{I}, \mathbb{J}}, \boldsymbol{x}_j) \leq d_{\boldsymbol{M}}(\boldsymbol{x}_{\text{test}} + \boldsymbol{\delta}_{\mathbb{I}, \mathbb{J}}, \boldsymbol{x}_i),
$$

$$
\forall j \in \mathbb{J}, \ \forall i \in \{i : y_i = y_{\text{test}}\} - \mathbb{I}. \tag{6}
$$

This minimization formulation enumerates all the $K$-size nearest neighbor set containing at most $k-1$ instances in the same class with the test instance, computes the minimal perturbation resulting in each $K$-nearest neighbor set, and takes the minimum of them.

Obviously, solving (6) exactly (enumerating all $(\mathbb{I}, \mathbb{J})$ pairs) has time complexity growing exponentially with $K$, and furthermore, a numerical solution cannot be incorporated into the training objective (5) since we need to write $\epsilon^*$ as a function of $\boldsymbol{M}$ for back-propagation. To address these issues, we resort to a lower bound of the optimal value of (6) rather than solving it exactly.

First, we consider a simple triplet problem: given vectors $\boldsymbol{x}^+, \boldsymbol{x}^-, \boldsymbol{x} \in \mathbb{R}^D$ and a positive semi-definite matrix $\boldsymbol{M} \in \mathbb{R}^{D \times D}$, find the minimum perturbation $\boldsymbol{\delta} \in \mathbb{R}^D$ on $\boldsymbol{x}$ such that $d_{\boldsymbol{M}}(\boldsymbol{x} + \boldsymbol{\delta}, \boldsymbol{x}^-) \leq d_{\boldsymbol{M}}(\boldsymbol{x} + \boldsymbol{\delta}, \boldsymbol{x}^+)$ holds. It could be formulated as the following optimization problem

$$
\min_{\boldsymbol{\delta}} \|\boldsymbol{\delta}\| \text{ s.t. } d_{\boldsymbol{M}}(\boldsymbol{x} + \boldsymbol{\delta}, \boldsymbol{x}^-) \leq d_{\boldsymbol{M}}(\boldsymbol{x} + \boldsymbol{\delta}, \boldsymbol{x}^+). \tag{7}
$$

Note that the constraint in (7) can be written as a linear form, so this is a convex quadratic programming problem with a linear constraint. We show that the optimal value of (7) can be expressed in closed form:

$$
\frac{[d_{\boldsymbol{M}}(\boldsymbol{x}, \boldsymbol{x}^-) - d_{\boldsymbol{M}}(\boldsymbol{x}, \boldsymbol{x}^+)]_+}{2\sqrt{(\boldsymbol{x}^+ - \boldsymbol{x}^-)^\top \boldsymbol{M}^\top \boldsymbol{M} (\boldsymbol{x}^+ - \boldsymbol{x}^-)}}, \tag{8}
$$

where $[\cdot]$ denotes $\max(\cdot, 0)$. The derivation for the optimal value is deferred to Appendix A. Note that if $\boldsymbol{M}$ is the identity matrix and $d_{\boldsymbol{M}}(\boldsymbol{x}, \boldsymbol{x}^-) > d_{\boldsymbol{M}}(\boldsymbol{x}, \boldsymbol{x}^+)$ strictly holds, the optimal value has a clear geometric meaning: it is the Euclidean distance from $\boldsymbol{x}$ to the bisection between $\boldsymbol{x}^+$ and $\boldsymbol{x}^-$.

For convenience, we define the function $\tilde{\epsilon} : \mathbb{R}^D \times \mathbb{R}^D \times \mathbb{R}^D \to \mathbb{R}$ as

$$
\tilde{\epsilon}(\boldsymbol{x}^+, \boldsymbol{x}^-, \boldsymbol{x}; \boldsymbol{M}) = \frac{d_{\boldsymbol{M}}(\boldsymbol{x}, \boldsymbol{x}^-) - d_{\boldsymbol{M}}(\boldsymbol{x}, \boldsymbol{x}^+)}{2\sqrt{(\boldsymbol{x}^+ - \boldsymbol{x}^-)^\top \boldsymbol{M}^\top \boldsymbol{M} (\boldsymbol{x}^+ - \boldsymbol{x}^-)}}. \tag{9}
$$

Then we could relax (6) further and have the following theorem:

**Theorem 1** (Robustness verification for Mahalanobis $K$-NN). *Given a Mahalanobis $K$-NN classifier parameterized by a neighbor parameter $K$, a training dataset $\mathbb{S}$ and a positive semi-definite matrix $\boldsymbol{M}$, for any instance $(\boldsymbol{x}_{test}, y_{test})$ we have*

$$
\epsilon^*(\boldsymbol{x}_{test}, y_{test}; \boldsymbol{M}) \geq \underset{j : y_j \neq y_{test}}{k\text{th} \min} \ \underset{i : y_i = y_{test}}{k\text{th} \max} \ \tilde{\epsilon}(\boldsymbol{x}_i, \boldsymbol{x}_j, \boldsymbol{x}_{test}; \boldsymbol{M}), \tag{10}
$$

*where $k\text{th} \max$ and $k\text{th} \min$ select the $k$-th maximum and $k$-th minimum respectively with $k = (K+1)/2$.*

The proof is deferred to Appendix B. In this way, we only need to compute $\tilde{\epsilon}(\boldsymbol{x}_i, \boldsymbol{x}_j, \boldsymbol{x}_{\text{test}})$ for each $i$ and $j$ in order to derive a lower bound of the minimal adversarial perturbation of Mahalanobis $K$-NN. It leads to an efficient algorithm to verify the robustness of Mahalanobis $K$-NN. The time complexity is $O(N^2)$ and independent of $K$. Note that any subset of $\{i : y_i = y_{\text{test}}\}$ also leads to a feasible lower

bound of the minimal adversarial perturbation and could improve computational efficiency, but the resulting lower bound is not necessarily as tight as (10). Therefore, in the experimental section, to evaluate certified robust errors as accurately as possible, we do not employ this strategy.

In the general multi-class case, the constraint of (6) is the necessary condition for successful attacks, rather than the necessary and sufficient condition. As a result, the optimal value of (6) is a lower bound of the minimal adversarial perturbation. Therefore, Theorem 1 also holds for the multi-class case. Based on this lower bound of $\epsilon^*$, we will derive the proposed ARML algorithm.

### 3.3 Training algorithm of adversarially robust metric learning

By replacing the $\epsilon^*$ in (5) with the lower bound derived in Theorem 1, we get a trainable objective function for adversarially robust metric learning:

$$\min_{\boldsymbol{G} \in \mathbb{R}^{D \times D}} \frac{1}{N} \sum_{t=1}^{N} \ell \left( \underset{j:y_j \neq y_t}{k\text{th}\min} \ \underset{i:i \neq t, y_i = y_t}{k\text{th}\max} \ \tilde{\epsilon}(\boldsymbol{x}_i, \boldsymbol{x}_j, \boldsymbol{x}_t; \boldsymbol{M}) \right) \quad \text{s.t. } \boldsymbol{M} = \boldsymbol{G}^\top \boldsymbol{G}. \quad (11)$$

Although (11) is trainable since $\tilde{\epsilon}$ is a function of $\boldsymbol{M}$, for large datasets it is time-consuming to run the inner min-max procedure. Furthermore, since what we really care is the generalization performance of the learned metric instead of the leave-one-out robust training error, it is unnecessary to compute the exact solution. Therefore, instead of computing the $k\text{th}\max$ and $k\text{th}\min$ exactly, we propose to sample positive and negative instances from the neighborhood of each training instance, which leads to the following formulation:

$$\min_{\boldsymbol{G} \in \mathbb{R}^{D \times D}} \frac{1}{N} \sum_{i=1}^{N} \ell \left( \tilde{\epsilon} \left( \text{randnear}_{\boldsymbol{M}}^+(\boldsymbol{x}_i), \text{randnear}_{\boldsymbol{M}}^-(\boldsymbol{x}_i), \boldsymbol{x}_i; \boldsymbol{M} \right) \right) \quad \text{s.t. } \boldsymbol{M} = \boldsymbol{G}^\top \boldsymbol{G}, \quad (12)$$

where $\text{randnear}_{\boldsymbol{M}}^+(\cdot)$ denotes a sampling procedure for an instance in the same class within $\boldsymbol{x}_i$'s neighborhood, and $\text{randnear}_{\boldsymbol{M}}^-(\cdot)$ denotes a sampling procedure for an instance in a different class, also within $\boldsymbol{x}_i$'s neighborhood, and the distances are measured by the Mahalanobis distance $d_{\boldsymbol{M}}$. In our implementation, we sample instances from a fixed number of nearest instances. As a result, the optimization formulation (12) approximately minimizes the certified robust error and improves computational efficient significantly.

Our *adversarially robust metric learning* (ARML) algorithm is shown in Algorithm 1. At every iteration, $\boldsymbol{G}$ is updated with the gradient of the loss function, while the calculations of $\text{randnear}_{\boldsymbol{M}}^+(\cdot)$ and $\text{randnear}_{\boldsymbol{M}}^-(\cdot)$ do not contribute to the gradient for the sake of efficient and stable computation.

---

**Algorithm 1:** Adversarially robust metric learning (ARML)

---

**Input:** Training data $\mathbb{S}$, number of epochs $T$.
**Output:** Positive semi-definite matrix $\boldsymbol{M}$.

1 Initialize $\boldsymbol{G}$ and $\boldsymbol{M}$ as identity matrices ;
2 **for** $t = 0 \ldots T - 1$ **do**
3      Update $\boldsymbol{G}$ with the gradient $\mathbb{E}_{(\boldsymbol{x},y) \in \mathbb{S}} \nabla_{\boldsymbol{G}} \ell \left( \tilde{\epsilon} \left( \text{randnear}_{\boldsymbol{M}}^+(\boldsymbol{x}), \text{randnear}_{\boldsymbol{M}}^-(\boldsymbol{x}), \boldsymbol{x}; \boldsymbol{G}^\top \boldsymbol{G} \right) \right)$;
4      Update $\boldsymbol{M}$ with the constraint $\boldsymbol{M} = \boldsymbol{G}^\top \boldsymbol{G}$;
5 **end**

---

### 3.4 Exact minimal adversarial perturbation of Mahalanobis 1-NN

In the special Mahalanobis 1-NN case, we will show a method to compute the exact minimal adversarial perturbation in a similar formulation to (6). However, this algorithm can only compute a numerical value of the minimal adversarial perturbation $\boldsymbol{\delta}^*$, so it cannot be used in training time. We will use this method to evaluate the robust error for the Mahalanobis 1-NN case in the experiments.

Computing the minimal adversarial perturbation $\epsilon_{\mathbb{S}}^*(\boldsymbol{x}_{\text{test}}, y_{\text{test}}; \boldsymbol{M})$ for Mahalanobis 1-NN classifier can be formulated as the following optimization problem:

$$\min_{j:y_j \neq y_{\text{test}}} \min_{\boldsymbol{\delta}_j} \|\boldsymbol{\delta}_j\| \quad \text{s.t. } d_{\boldsymbol{M}}(\boldsymbol{x}_{\text{test}} + \boldsymbol{\delta}_j, \boldsymbol{x}_j) \leq d_{\boldsymbol{M}}(\boldsymbol{x}_{\text{test}} + \boldsymbol{\delta}_j, \boldsymbol{x}_i), \ \forall i : y_i = y_{\text{test}}. \quad (13)$$

Interestingly and not surprisingly, it is a special case of (6) where we have $K = 1$ and $k = (K + 1)/2 = 1$, and hence $\mathbb{I}$ is an empty set, and $\mathbb{J}$ has only one element. The formulation of (13) is equivalent to considering each $\boldsymbol{x}_j$ in a different class from $y_{\text{test}}$ and computing the minimum perturbation needed for making $\boldsymbol{x}_{\text{test}}$ closer to $\boldsymbol{x}_j$ than all the training instances in the same class with $y_{\text{test}}$, i.e., $\boldsymbol{x}_i$s,. It is noteworthy that the constraint of (13) could be equivalently written as

$$(\boldsymbol{x}_i - \boldsymbol{x}_j)^\top \boldsymbol{M} \boldsymbol{\delta} \leq \frac{1}{2} \left( d_{\boldsymbol{M}}(\boldsymbol{x}_{\text{test}}, \boldsymbol{x}_i) - d_{\boldsymbol{M}}(\boldsymbol{x}_{\text{test}}, \boldsymbol{x}_j) \right), \ \forall i : y_i = y_{\text{test}}, \tag{14}$$

which are all affine functions. Therefore, the inner minimization is a convex quadratic programming problem and could be solved in polynomial time [25]. As a result, it leads to a naive polynomial-time algorithm for finding the minimal adversarial perturbation of Mahalanobis 1-NN: solve all the inner convex quadratic programming problems and then select the minimum of them.

Instead, we propose a much more efficient method to solve (13). The main idea is to compute a lower bound for each inner minimization problem first, and with these lower bounds, we could screen most of the inner minimization problems safely without the need of solving them exactly. This method is an extension of our previous work [45], where only the Euclidean distance is taken into consideration. See Algorithm 2 in Appendix C for details and this algorithm is used for computing certified robust errors of Mahalanobis 1-NN in the experimental section.

# 4 Experiments

We compare the proposed ARML (Adversarial Robust Metric Learning) method with the following baselines:

- Euclidean: uses the Euclidean distance directly without learning any metric;
- Neighbourhood components analysis (NCA) [17]: maximizes a stochastic variant of the leave-one-out nearest neighbors score on the training set.
- Large margin nearest neighbor (LMNN) [48]: keeps close nearest neighbors from the same class, while keeps instances from different classes separated by a large margin.
- Information Theoretic Metric Learning (ITML) [13]: minimizes the log-determinant divergence with similarity and dissimilarity constraints.
- Local Fisher Discriminant Analysis (LFDA) [42]: a modified version of linear discriminant analysis by rewriting scatter matrices in a pairwise manner.

For evaluation, we use six public datasets on which metric learning methods perform favorably in terms of clean errors, including four small or medium-sized datasets [5]: Splice, Pendigits, Satimage and USPS, and two image datasets MNIST [31] and Fashion-MNIST [51], which are wildly used for robustness verification for neural networks. For the proposed method, we use the same hyperparameters for all the datasets (see Appendix D for the dataset statistics, more details of the experimental setting, and hyperparameter sensitivity analysis).

## 4.1 Mahalanobis 1-NN

Certified robust errors of Mahalanobis 1-NN with respect to different perturbation radii are shown in Table 1. It should be noted that these radii are only used to show the experimental results, and they are not hyperparameters. In this Mahalanobis 1-NN case, the proposed algorithm in Algorithm 2, which solves (13), can compute the exact minimal adversarial perturbation for each instance, so the values we get in Table 1 are both (optimal) certified robust errors and (optimal) empirical robust errors (attack errors). Also, note that when the radius is 0, the resulting certified robust error is equivalent to the clean error on the unperturbed test set.

We have three main observations from the experimental results. First, although NCA and LMNN achieve better clean errors (at the radius 0) than Euclidean in most datasets, they are less robust to adversarial perturbations than Euclidean (except the Splice dataset, on which Euclidean performs overly poorly in terms of clean errors and then has a large robust errors accordingly). Both NCA and LMNN suffer from the trade-off between the clean error and the certified robust error. Second, ARML performs competitively with NCA and LMNN in terms of clean errors (achieves the best on 4/6 of the datasets). Third and the most importantly, ARML is much more robust than all the other methods in terms of certified robust errors for all perturbation radii.

Table 1: Certified robust errors of Mahalanobis 1-NN. The best (minimum) certified robust errors among all methods are in bold. Note that the certified robust errors of 1-NN are also the optimal empirical robust errors (attack errors), and these robust errors at the radius 0 are also the clean errors.

| | $\ell_2$-radius | 0.000 | 0.500 | 1.000 | 1.500 | 2.000 | 2.500 |
|---|---|---|---|---|---|---|---|
| **MNIST** | Euclidean | 0.033 | 0.112 | 0.274 | 0.521 | 0.788 | 0.945 |
| | NCA | 0.025 | 0.140 | 0.452 | 0.839 | 0.977 | 1.000 |
| | LMNN | 0.032 | 0.641 | 0.999 | 1.000 | 1.000 | 1.000 |
| | ITML | 0.073 | 0.571 | 0.928 | 1.000 | 1.000 | 1.000 |
| | LFDA | 0.152 | 1.000 | 1.000 | 1.000 | 1.000 | 1.000 |
| | ARML (Ours) | **0.024** | **0.089** | **0.222** | **0.455** | **0.757** | **0.924** |
| | $\ell_2$-radius | 0.000 | 0.500 | 1.000 | 1.500 | 2.000 | 2.500 |
| **Fashion-MNIST** | Euclidean | 0.145 | 0.381 | 0.606 | 0.790 | 0.879 | 0.943 |
| | NCA | **0.116** | 0.538 | 0.834 | 0.950 | 0.998 | 1.000 |
| | LMNN | 0.142 | 0.756 | 0.991 | 1.000 | 1.000 | 1.000 |
| | ITML | 0.163 | 0.672 | 0.929 | 0.998 | 1.000 | 1.000 |
| | LFDA | 0.211 | 1.000 | 1.000 | 1.000 | 1.000 | 1.000 |
| | ARML (Ours) | 0.127 | **0.348** | **0.568** | **0.763** | **0.859** | **0.928** |
| | $\ell_2$-radius | 0.000 | 0.100 | 0.200 | 0.300 | 0.400 | 0.500 |
| **Splice** | Euclidean | 0.320 | 0.513 | 0.677 | 0.800 | 0.854 | 0.880 |
| | NCA | **0.130** | 0.252 | 0.404 | 0.584 | 0.733 | 0.836 |
| | LMNN | 0.190 | 0.345 | 0.533 | 0.697 | 0.814 | 0.874 |
| | ITML | 0.306 | 0.488 | 0.679 | 0.809 | 0.862 | 0.882 |
| | LFDA | 0.264 | 0.434 | 0.605 | 0.760 | 0.845 | 0.872 |
| | ARML (Ours) | **0.130** | **0.233** | **0.370** | **0.526** | **0.652** | **0.758** |
| | $\ell_2$-radius | 0.000 | 0.100 | 0.200 | 0.300 | 0.400 | 0.500 |
| **Pendigits** | Euclidean | 0.032 | 0.119 | 0.347 | 0.606 | 0.829 | 0.969 |
| | NCA | 0.034 | 0.202 | 0.586 | 0.911 | 0.997 | 1.000 |
| | LMNN | 0.029 | 0.183 | 0.570 | 0.912 | 0.995 | 0.999 |
| | ITML | 0.049 | 0.308 | 0.794 | 0.991 | 1.000 | 1.000 |
| | LFDA | 0.042 | 0.236 | 0.603 | 0.912 | 0.998 | 1.000 |
| | ARML (Ours) | **0.028** | **0.115** | **0.344** | **0.598** | **0.823** | **0.967** |
| | $\ell_2$-radius | 0.000 | 0.150 | 0.300 | 0.450 | 0.600 | 0.750 |
| **Satimage** | Euclidean | 0.108 | 0.642 | 0.864 | 0.905 | 0.928 | 0.951 |
| | NCA | 0.103 | 0.710 | 0.885 | 0.915 | 0.940 | 0.963 |
| | LMNN | **0.092** | 0.665 | 0.871 | 0.912 | 0.944 | 0.969 |
| | ITML | 0.127 | 0.807 | 0.979 | 1.000 | 1.000 | 1.000 |
| | LFDA | 0.125 | 0.836 | 0.919 | 0.956 | 0.992 | 1.000 |
| | ARML (Ours) | 0.095 | **0.605** | **0.839** | **0.899** | **0.920** | **0.946** |
| | $\ell_2$-radius | 0.000 | 0.500 | 1.000 | 1.500 | 2.000 | 2.500 |
| **USPS** | Euclidean | 0.045 | 0.224 | 0.585 | 0.864 | **0.970** | **0.999** |
| | NCA | 0.056 | 0.384 | 0.888 | 0.987 | 1.000 | 1.000 |
| | LMNN | 0.046 | 0.825 | 1.000 | 1.000 | 1.000 | 1.000 |
| | ITML | 0.060 | 0.720 | 0.999 | 1.000 | 1.000 | 1.000 |
| | LFDA | 0.098 | 1.000 | 1.000 | 1.000 | 1.000 | 1.000 |
| | ARML (Ours) | **0.043** | **0.204** | **0.565** | **0.857** | **0.970** | **0.999** |

## 4.2 Mahalanobis $K$-NN

For $K$-NN models, it is intractable to compute the exact minimal adversarial perturbation, so we report both certified robust errors and empirical robust errors (attack errors). We set $K = 11$ for all the experiments. The certified robust error can be computed by Theorem 1, which works for any Mahalanobis distance. On the other hand, we also conduct adversarial attacks to these models to derive the empirical robust error — the lower bounds of the certified robust errors — via a hard-label black-box attack method, i.e., the Boundary Attack [3]. Different from the 1-NN case, since both adversarial attack and robustness verification are not optimal, there will be a gap between the two kinds of robust errors. These results are shown in Table 2. Note that these empirical robust errors at the radius 0 are also the clean errors.

The three observations of Mahalanobis 1-NN also hold for the $K$-NN case: NCA and LMNN have improved clean errors (empirical robust errors at the radius 0) but this often comes with degraded robust errors compared with the Euclidean distance, while ARML achieves good robust errors as well

Table 2: Certified robust errors (left) and empirical robust errors (right) of Mahalanobis $K$-NN. The best (minimum) robust errors among all methods are in bold. The empirical robust errors at the radius 0 are also the clean errors.

| | | Certified robust errors | | | | | | Empirical robust errors | | | | | |
|---|---|---|---|---|---|---|---|---|---|---|---|---|---|
| | $\ell_2$-radius | 0.000 | 0.500 | 1.000 | 1.500 | 2.000 | 2.500 | 0.000 | 0.500 | 1.000 | 1.500 | 2.000 | 2.500 |
| MNIST | Euclidean | 0.038 | 0.134 | 0.360 | 0.618 | 0.814 | 0.975 | 0.031 | 0.063 | 0.104 | 0.155 | 0.204 | 0.262 |
| | NCA | **0.030** | 0.175 | 0.528 | 0.870 | 0.986 | 1.000 | **0.027** | 0.063 | 0.120 | 0.216 | 0.330 | 0.535 |
| | LMNN | 0.040 | 0.669 | 1.000 | 1.000 | 1.000 | 1.000 | 0.036 | 0.121 | 0.336 | 0.775 | 0.972 | 1.000 |
| | ITML | 0.106 | 0.731 | 0.943 | 1.000 | 1.000 | 1.000 | 0.084 | 0.218 | 0.355 | 0.510 | 0.669 | 0.844 |
| | LFDA | 0.237 | 1.000 | 1.000 | 1.000 | 1.000 | 1.000 | 0.215 | 1.000 | 1.000 | 1.000 | 1.000 | 1.000 |
| | ARML (Ours) | 0.034 | **0.101** | **0.276** | **0.537** | **0.760** | **0.951** | 0.032 | **0.055** | **0.077** | **0.109** | **0.160** | **0.213** |
| | $\ell_2$-radius | 0.000 | 0.500 | 1.000 | 1.500 | 2.000 | 2.500 | 0.000 | 0.500 | 1.000 | 1.500 | 2.000 | 2.500 |
| Fashion-MNIST | Euclidean | 0.160 | 0.420 | 0.650 | 0.800 | 0.895 | 0.946 | 0.143 | 0.227 | 0.298 | 0.360 | 0.420 | 0.489 |
| | NCA | **0.144** | 0.557 | 0.832 | 0.946 | 1.000 | 1.000 | **0.121** | 0.232 | 0.343 | 0.483 | 0.624 | 0.780 |
| | LMNN | 0.158 | 0.792 | 0.991 | 1.000 | 1.000 | 1.000 | 0.140 | 0.364 | 0.572 | 0.846 | 0.983 | 0.999 |
| | ITML | 0.236 | 0.784 | 0.949 | 1.000 | 1.000 | 1.000 | 0.209 | 0.460 | 0.692 | 0.892 | 0.978 | 1.000 |
| | LFDA | 0.291 | 1.000 | 1.000 | 1.000 | 1.000 | 1.000 | 0.263 | 0.870 | 0.951 | 0.975 | 0.988 | 0.995 |
| | ARML (Ours) | 0.152 | **0.371** | **0.589** | **0.755** | **0.856** | **0.924** | 0.134 | **0.202** | **0.274** | **0.344** | **0.403** | **0.487** |
| | $\ell_2$-radius | 0.000 | 0.100 | 0.200 | 0.300 | 0.400 | 0.500 | 0.000 | 0.100 | 0.200 | 0.300 | 0.400 | 0.500 |
| Splice | Euclidean | 0.333 | 0.558 | 0.826 | 0.965 | 0.988 | 0.996 | 0.306 | 0.431 | 0.526 | 0.608 | 0.676 | 0.743 |
| | NCA | **0.103** | **0.209** | 0.415 | 0.659 | 0.824 | 0.921 | **0.103** | 0.173 | 0.274 | 0.414 | 0.570 | 0.684 |
| | LMNN | 0.149 | 0.332 | 0.630 | 0.851 | 0.969 | 0.994 | 0.149 | 0.241 | 0.357 | 0.492 | 0.621 | 0.722 |
| | ITML | 0.279 | 0.571 | 0.843 | 0.974 | 0.995 | 0.997 | 0.279 | 0.423 | 0.525 | 0.603 | 0.675 | 0.751 |
| | LFDA | 0.242 | 0.471 | 0.705 | 0.906 | 0.987 | 0.997 | 0.242 | 0.371 | 0.466 | 0.553 | 0.637 | 0.737 |
| | ARML (Ours) | 0.128 | 0.221 | **0.345** | **0.509** | **0.666** | **0.819** | 0.128 | 0.196 | **0.273** | **0.380** | **0.497** | **0.639** |
| | $\ell_2$-radius | 0.000 | 0.100 | 0.200 | 0.300 | 0.400 | 0.500 | 0.000 | 0.100 | 0.200 | 0.300 | 0.400 | 0.500 |
| Pendigits | Euclidean | 0.039 | 0.126 | 0.316 | 0.577 | 0.784 | 0.937 | 0.036 | 0.085 | 0.155 | 0.248 | 0.371 | 0.528 |
| | NCA | 0.038 | 0.196 | 0.607 | 0.884 | 0.997 | 1.000 | 0.038 | 0.103 | 0.246 | 0.428 | 0.637 | 0.804 |
| | LMNN | 0.034 | 0.180 | 0.568 | 0.898 | 0.993 | 0.999 | **0.030** | 0.096 | 0.246 | 0.462 | 0.681 | 0.862 |
| | ITML | 0.060 | 0.334 | 0.773 | 0.987 | 1.000 | 1.000 | 0.060 | 0.149 | 0.343 | 0.616 | 0.814 | 0.926 |
| | LFDA | 0.047 | 0.228 | 0.595 | 0.904 | 1.000 | 1.000 | 0.043 | 0.104 | 0.248 | 0.490 | 0.705 | 0.842 |
| | ARML (Ours) | **0.035** | **0.114** | **0.308** | **0.568** | **0.780** | **0.937** | 0.034 | **0.078** | **0.138** | **0.235** | **0.368** | **0.516** |
| | $\ell_2$-radius | 0.000 | 0.150 | 0.300 | 0.450 | 0.600 | 0.750 | 0.000 | 0.150 | 0.300 | 0.450 | 0.600 | 0.750 |
| Satimage | Euclidean | **0.101** | 0.579 | 0.842 | 0.899 | 0.927 | 0.948 | 0.091 | 0.237 | 0.482 | 0.682 | **0.816** | 0.897 |
| | NCA | 0.117 | 0.670 | 0.886 | 0.915 | 0.936 | 0.961 | 0.101 | 0.297 | 0.564 | 0.746 | 0.876 | 0.931 |
| | LMNN | 0.105 | 0.613 | 0.855 | 0.914 | 0.944 | 0.961 | **0.090** | 0.269 | 0.548 | 0.737 | 0.855 | 0.910 |
| | ITML | 0.130 | 0.768 | 0.959 | 1.000 | 1.000 | 1.000 | 0.109 | 0.411 | 0.757 | 0.939 | 0.990 | 1.000 |
| | LFDA | 0.128 | 0.779 | 0.904 | 0.958 | 0.995 | 1.000 | 0.112 | 0.389 | 0.673 | 0.860 | 0.950 | 0.986 |
| | ARML (Ours) | 0.103 | **0.540** | **0.824** | **0.898** | **0.920** | **0.943** | 0.092 | **0.228** | **0.464** | **0.668** | 0.817 | **0.896** |
| | $\ell_2$-radius | 0.000 | 0.500 | 1.000 | 1.500 | 2.000 | 2.500 | 0.000 | 0.500 | 1.000 | 1.500 | 2.000 | 2.500 |
| USPS | Euclidean | 0.063 | 0.239 | 0.586 | 0.888 | 0.977 | 1.000 | 0.058 | 0.125 | 0.211 | 0.365 | 0.612 | **0.751** |
| | NCA | 0.072 | 0.367 | 0.903 | 0.986 | 1.000 | 1.000 | 0.063 | 0.158 | 0.365 | 0.686 | 0.899 | 0.980 |
| | LMNN | 0.062 | 0.856 | 1.000 | 1.000 | 1.000 | 1.000 | 0.055 | 0.359 | 0.890 | 0.999 | 1.000 | 1.000 |
| | ITML | 0.082 | 0.696 | 0.999 | 1.000 | 1.000 | 1.000 | 0.072 | 0.273 | 0.708 | 0.987 | 1.000 | 1.000 |
| | LFDA | 0.134 | 1.000 | 1.000 | 1.000 | 1.000 | 1.000 | 0.118 | 0.996 | 1.000 | 1.000 | 1.000 | 1.000 |
| | ARML (Ours) | **0.057** | **0.203** | **0.527** | **0.867** | **0.971** | **0.997** | **0.053** | **0.118** | **0.209** | **0.344** | **0.572** | 0.785 |

as clean errors. The results suggest that ARML is more robust both *provably* (in terms of the certified robust error) and empirically (in terms of the empirical robust error).

## 4.3 Comparison with neural networks

We compare Mahalanobis 1-NN classifiers with neural networks, including ordinary neural networks certified by the robustness verification method CROWN [55] and randomized-smoothing neural networks (with smoothness parameters 0.2 and 1) [12]. The results are shown in Figure 2. It is shown that randomized smoothing encounters a trade-off between clean and robust errors, whereas ARML does not sacrifice clean errors compared with previous metric learning methods.

## 4.4 Computational cost

In general, computational cost is not an issue for ARML. The average runtime (of 5 trials) of LMNN, ITML, NCA and ARML, are 66.4s, 95.2s, 480.9s and 146.1s respectively on USPS for 100 iterations in total, where to make the comparison fair, all of the methods are run on CPU (Xeon(R) E5-2620 v4 @2.10GHz). In fact, ARML is highly parallelizable and our implementation also supports GPU with

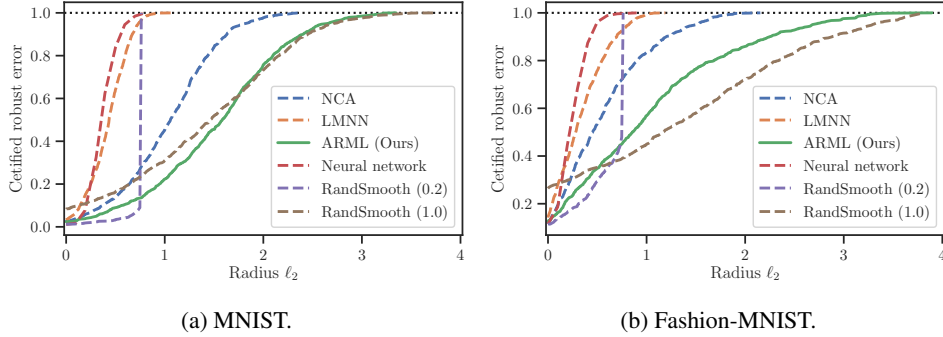

|                | (a) MNIST. | (b) Fashion-MNIST. |
|----------------|------------|--------------------|

Figure 2: Certified robust errors comparing neural networks.

the PyTorch library [41]: when running on GPU (one Nvidia TITAN Xp), the average runtime of ARML is only 10.6s.

## 5 Related work

**Metric learning**   Metric learning aims to learn a new distance metric using supervision information concerning the learned distance [26]. In this paper, we mainly focus on the linear metric learning: the learned distance is the squared Euclidean distance after applying a linear transformation to instances globally, i.e., the Mahalanobis distance [17, 13, 48, 22, 42]. It is noteworthy that there are also nonlinear models for metric learning, such as kernelized metric learning [28, 6], local metric learning [16, 47] and deep metric learning [11, 37]. Robustness verification for nonlinear metric learning and provably robust non-linear metric learning (certified defense for non-linear metric learning) would be an interesting future work.

**Adversarial robustness of neural networks**   Empirical defense is usually able to learn a classifier which is robust to some specific adversarial attacks [29, 33], but has no guarantee for the robustness to other stronger (or unknown) adversarial attacks [4, 1]. In contrast, certified defense provides a guarantee that no adversarial examples exist within a certain input region [50, 12, 55]. The basic idea of these certified defense methods is to minimize the certified robust error on the training data. However, all these certified defense methods for neural networks rely on the assumption of smoothness of the classifier, and hence could not be applied to the nearest neighbor classifiers.

**Adversarial robustness and metric learning**   Some papers introduce the adversarial framework as a mining strategy, aiming to improve classification accuracy of metric learning [8, 15, 56], and others employ some metric learning loss functions as regularization to improve empirical robustness of deep learning models [34]. However, few papers investigate adversarial robustness, especially certified adversarial robustness of metric learning models themselves.

**Adversarial robustness of nearest neighbor classifiers**   Most works about adversarial robustness of $K$-NN focus on adversarial attack. Some papers propose to attack a differentiable substitute of $K$-NN [36, 32, 38], and others formalize the attack as a list of quadratic programming problems or linear programming problems [45, 53]. As far as we know, there is only one paper (our previous work) considering robustness verification for $K$-NN, but they only consider the Euclidean distance, and no certified defense method is proposed [45]. In contrast, we propose the first adversarial verification method and the first certified defense (provably robust learning) method for Mahalanobis $K$-NN.

## 6 Conclusion

We propose a novel metric learning method named ARML to obtain a robust Mahalanobis distance that can be robust to adversarial input perturbations. Experiments show that the proposed method can simultaneously improve clean classification accuracy and adversarial robustness (in terms of both certified robust errors and empirical robust errors) compared with existing metric learning algorithms.

## Broader impact

In this work, we study the problem of adversarial robustness of metric learning. Adversarial robustness, especially robustness verification, is very important when deploying machine learning models into real-world systems. A potential risk is the research on adversarial attack, while understanding adversarial attack is a necessary step towards developing provably robust models. In general, this work does not involve specific applications and ethical issues.

## Acknowledgement

This work was jointly supported by NSFC 61673201, NSFC 61921006, NSF IIS-1901527, NSF IIS-2008173, ARL-0011469453, and Facebook.

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
