[Supplementary Material]

# A  Optimal value of triplet problem

The triplet problem is formalized as below:

$$\min_{\boldsymbol{\delta}} \|\boldsymbol{\delta}\| \text{ s.t. } d_M(\boldsymbol{x} + \boldsymbol{\delta}, \boldsymbol{x}^-) \leq d_M(\boldsymbol{x} + \boldsymbol{\delta}, \boldsymbol{x}^+). \tag{15}$$

It is equivalent to the optimization

$$\min_{\boldsymbol{\delta}} \boldsymbol{\delta}^\top \boldsymbol{\delta} \text{ s.t. } \boldsymbol{a}^\top \boldsymbol{\delta} \leq b, \tag{16}$$

where we have

$$\boldsymbol{a} = M\left(\boldsymbol{x}^+ - \boldsymbol{x}^-\right), \tag{17}$$

$$b = \frac{1}{2}\left(d_M(\boldsymbol{x}, \boldsymbol{x}^+) - d_M(\boldsymbol{x}, \boldsymbol{x}^-)\right). \tag{18}$$

The dual function is

$$g(\lambda) = \inf_{\boldsymbol{\delta}} \ \boldsymbol{\delta}^\top \boldsymbol{\delta} + \lambda(\boldsymbol{a}^\top \boldsymbol{\delta} - b) \tag{19}$$

$$= -\frac{1}{4}\boldsymbol{a}^\top \boldsymbol{a}\lambda^2 - b\lambda, \tag{20}$$

where inf holds for $\boldsymbol{\delta} = -\lambda \boldsymbol{a}/2$. Then the dual problem is

$$\max_{\lambda \geq 0} \ -\frac{1}{4}\boldsymbol{a}^\top \boldsymbol{a}\lambda^2 - b\lambda. \tag{21}$$

The optimal point is

$$\left[-\frac{2b}{\boldsymbol{a}^\top \boldsymbol{a}}\right]_+, \tag{22}$$

and the optimal value is

$$\begin{cases} 0 & \text{if } b \geq 0 \\ \frac{b^2}{\boldsymbol{a}^\top \boldsymbol{a}} & \text{otherwise.} \end{cases} \tag{23}$$

By the Slater's condition, if $\boldsymbol{x}^+ \neq \boldsymbol{x}^-$ holds, we have the strong duality. Therefore, the optimal value of (15) is

$$\left[\frac{-b}{\sqrt{\boldsymbol{a}^\top \boldsymbol{a}}}\right]_+ = \frac{[d_M(\boldsymbol{x}, \boldsymbol{x}^-) - d_M(\boldsymbol{x}, \boldsymbol{x}^+)]_+}{2\sqrt{(\boldsymbol{x}^+ - \boldsymbol{x}^-)^\top M^\top M (\boldsymbol{x}^+ - \boldsymbol{x}^-)}}. \tag{24}$$

In fact, it is easy to verify that even if $\boldsymbol{x}^+ = \boldsymbol{x}^-$ obtains, the optimal value also holds.

# B  Proof of Theorem 1

*Proof.* Let $\epsilon^{(\mathbb{I},\mathbb{J})}$ denote the optimal value of the inner minimization problem of (6). By relaxing the constraint via replacing the universal quantifier, we have

$$\epsilon^{(\mathbb{I},\mathbb{J})} \geq \max_{i \in \{i:y_i = y_{\text{test}}\} - \mathbb{I}, j \in \mathbb{J}} \tilde{\epsilon}(\boldsymbol{x}_i, \boldsymbol{x}_j, \boldsymbol{x}_{\text{test}}; M). \tag{25}$$

Substitute it in (6) and then we have

$$\epsilon^* \geq \min_{\mathbb{I},\mathbb{J}} \epsilon^{(\mathbb{I},\mathbb{J})} \tag{26}$$

$$\geq \min_{\mathbb{I},\mathbb{J}} \max_{i \in \{i:y_i = y_{\text{test}}\} - \mathbb{I}, \ j \in \mathbb{J}} \tilde{\epsilon}(\boldsymbol{x}_i^+, \boldsymbol{x}_j^-, \boldsymbol{x}_{\text{test}}) \tag{27}$$

$$\geq \min_{\mathbb{I},\mathbb{J}} \max_{j \in \mathbb{J}} \max_{i \in [\{i:y_i = y_{\text{test}}\} - \mathbb{I}} \tilde{\epsilon}(\boldsymbol{x}_i^+, \boldsymbol{x}_j^-, \boldsymbol{x}_{\text{test}}) \tag{28}$$

$$\geq \min_{\mathbb{I},\mathbb{J}} \max_{j \in \mathbb{J}} \ k\text{th} \max_{i \in \{i:y_i = y_{\text{test}}\}} \tilde{\epsilon}(\boldsymbol{x}_i^+, \boldsymbol{x}_j^-, \boldsymbol{x}_{\text{test}}) \tag{29}$$

$$\geq \min_{\mathbb{I},\mathbb{J}} \ k\text{th} \min_{j \in \{j:y_j \neq y_{\text{test}}\}} \ k\text{th} \max_{i \in \{i:y_i = y_{\text{test}}\}} \tilde{\epsilon}(\boldsymbol{x}_i^+, \boldsymbol{x}_j^-, \boldsymbol{x}_{\text{test}}) \tag{30}$$

$$= \ k\text{th} \min_{j \in \{j:y_j \neq y_{\text{test}}\}} \ k\text{th} \max_{i \in \{i:y_i = y_{\text{test}}\}} \tilde{\epsilon}(\boldsymbol{x}_i^+, \boldsymbol{x}_j^-, \boldsymbol{x}_{\text{test}}) \tag{31}$$

$\square$

## C  Details of computing exact minimal adversarial perturbation of Mahalanobis 1-NN

The overall algorithm is displayed in Algorithm 2. We denote $\epsilon^{(j)}$ as the optimal value of the inner minimization problem with respect to $j$, and denote $\underline{\epsilon}^{(j)}$ as its lower bound. We first sort the subproblems according to the ascending order of $\|\boldsymbol{x}_j - \boldsymbol{x}_{\text{test}}\|$ for $\{j : y_j \neq y_{\text{test}}\}$. For every subproblem, we compute the lower bound of its optimal value. If the optimal value is too large, we just screen the subproblem safely without solving it exactly.

---

**Algorithm 2:** Computing the minimal adversarial perturbation for Mahalanobis 1-NN

---

**Input:** Test instance $(\boldsymbol{x}_{\text{test}}, y_{\text{test}})$, dataset $\mathbb{S} = \{(\boldsymbol{x}_i, y_i)\}_{i=1}^N$.
**Output:** Perturbation norm $\epsilon$.
1  Initialize $\epsilon = \infty$ ;
2  Sort $\{j : y_j \neq y_{\text{test}}\}$ by the ascending order of $d_{\boldsymbol{M}}(\boldsymbol{x}_{\text{test}}, \boldsymbol{x}_j)$;
3  **for** $j : y_j \neq y_{test}$ *according to the ascending order* **do**
4  $\quad$ Compute a lower bound $\underline{\epsilon}^{(j)}$ of the inner minimization corresponding to $j$;
5  $\quad$ **if** $\underline{\epsilon} < \epsilon$ **then**
6  $\quad\quad$ Solve the inner minimization problem exactly via the greedy coordinate ascent method
$\quad\quad$ and derive the optimal value $\epsilon^{(j)}$;
7  $\quad\quad$ **if** $\epsilon^{(j)} < \epsilon$ **then**
8  $\quad\quad\quad$ $\epsilon = \epsilon^{(j)}$
9  $\quad\quad$ **end**
10 $\quad$ **end**
11 **end**

---

### C.1  Greedy coordinate ascent (descent)

For the subproblem we have to solve exactly, we employ the greedy coordinate ascent method. Note that the inner minimization problem of (13) is a convex quadratic programming problem. We solve the problem by dealing with its dual formulation. The greedy coordinate ascent method is used because the optimal dual variables are very sparse. The algorithm is shown in Algorithm 3. At every iteration, only one dual variable is updated.

---

**Algorithm 3:** Greedy coordinate descent for QP: $\min_{\boldsymbol{x} \geq 0} \frac{1}{2}\boldsymbol{x}^\top \boldsymbol{P}\boldsymbol{x} + \boldsymbol{q}^\top \boldsymbol{x}$

---

**Input:** $\boldsymbol{P}, \boldsymbol{q}, \epsilon, T$.
1  $\boldsymbol{x} \leftarrow \boldsymbol{0}, \boldsymbol{g} \leftarrow \boldsymbol{P}\boldsymbol{x} + \boldsymbol{q}$;
2  **for** $t = 0$ *to* $T - 1$ **do**
3  $\quad$ $\forall i, y_i \leftarrow \max\left(x_i - \frac{g_i}{p_{i,i}}, 0\right) - x_i$;
4  $\quad$ $i^* \leftarrow \arg\max_i |y_i|$ ; // choose a coordinate
5  $\quad$ **if then**
6  $\quad\quad$ break;
7  $\quad$ **end**
8  $\quad$ $x_{i^*} \leftarrow x_{i^*} + y_{i^*}$ ; // update the solution
9  $\quad$ $\boldsymbol{g} \leftarrow \boldsymbol{g} + y_{i^*}\boldsymbol{p}_{i^*}$ ; // update the gradient
10 **end**
**Output:** $\boldsymbol{x}$.

---

### C.2  Lower bound of inner minimization problem

The following theorem is dependent on the solution of the triplet problem.

**Theorem 2.** *The optimal value $\epsilon^{(j)}$ of the inner minimization of* (13) *with respect to $j$ is lower bounded as*

$$\epsilon^{(j)} \geq \max_{i:y_i=y_{test}} \left[\tilde{\epsilon}(\boldsymbol{x}_i, \boldsymbol{x}_j, \boldsymbol{x}_{test}; \boldsymbol{M})\right]_+ . \tag{32}$$

*Proof.* Relaxing the constraint of (13) by means of replacing the universal quantifier, we know $\epsilon^{(j)}$ is lower bounded by the optimal value of the following optimization problem

$$\max_{i:y_i=y_{\text{test}}} \min_{\boldsymbol{\delta}_{i,j}} \|\boldsymbol{\delta}_{i,j}\| \tag{33}$$

$$\text{s.t. } d_{\boldsymbol{M}}(\boldsymbol{x}_{\text{test}} + \boldsymbol{\delta}_{i,j}, \boldsymbol{x}_j) \leq d_{\boldsymbol{M}}(\boldsymbol{x}_{\text{test}} + \boldsymbol{\delta}_{i,j}, \boldsymbol{x}_i). \tag{34}$$

Obviously, the optimal value of the inner problem is $[\tilde{\epsilon}(\boldsymbol{x}_i, \boldsymbol{x}_j, \boldsymbol{x}_{\text{test}}; \boldsymbol{M})]_+$. $\square$

In this way, we could derive a lower bound of the optimal value in closed form.

## D   Experimental details

**Datasets**   Dataset statistics and test errors (on all test instances) of Euclidean $K$-NN with $K = 11$ are shown in Table 3. All training data are used to learn metrics, and 1,000 instances are randomly sampled to compute certified robust errors.

Table 3: Dataset statisitcs.

| | # features | # classes | # train | # test | 1-NN test error | $K$-NN test error |
|---|---|---|---|---|---|---|
| MNIST | 784 | 10 | 60,000 | 10,000 | 0.031 | 0.033 |
| Fashion-MNIST | 784 | 10 | 60,000 | 10,000 | 0.150 | 0.150 |
| Splice | 60 | 2 | 1,000 | 2,175 | 0.295 | 0.291 |
| Pendigits | 16 | 10 | 7,494 | 3,498 | 0.023 | 0.027 |
| Satimage | 36 | 6 | 4,435 | 2,000 | 0.112 | 0.106 |
| USPS | 256 | 10 | 7,291 | 2,007 | 0.049 | 0.060 |

**Hyperparameters**   Hyperparameters of our ARML algorithm are fixed across all datasets. Specifically, the size of the neighborhood where $\text{randnear}^+$ and $\text{randnear}^-$ sample random instances is 10. In other words, at every iteration, we sample one instance from the nearest 10 instances in the same class with the test instance, and sample one instance from the nearest 10 instances in the different classes from the test instance. We employ the Adam algorithm [24] to update parameters with gradients and the parameters is in the default setting (learning rate: 0.001, betas: $(0.9, 0.999)$). The number of epochs is 1,000. The loss function is the negative loss.

**Hyperparameter sensitivity**   We investigate the sensitivity of the size of neighborhood used for $\text{randnear}^+$ and $\text{randnear}^-$. We plot the robust error curves against the $\ell_2$ radius for different neighborhood sizes in Figure 3 and Figure 4. It suggests that ARML is not very sensitive to this hyperparameter in terms of certified and empirical robust errors.

(a) 1-NN certified robust error.        (b) $K$-NN certified robust error.        (c) $K$-NN empirical robust error.

Figure 3: Sensitivity to neighborhood size on Splice.

(a) 1-NN certified robust error.      (b) $K$-NN certified robust error.      (c) $K$-NN empirical robust error.

Figure 4: Sensitivity to neighborhood size on Satimage.

**Implementations of NCA, LMNN, ITML and LFDA**    We use the implementations of the metric-learn library [14] for NCA, LMNN, ITML and LFDA. Similar to ARML, hyperparameters are fixed across all datasets and are in the default setting. In particular, the maximum numbers of iterations for NCA, LMNN and ITML are 1,00, 1000 and 1,000 respectively.