[Reviews · NeurIPS 2020]

Review 1

Summary and Contributions: The authors proposed a metric learning algorithm to find a Mahalanobis distance that is robust against adversarial perturbation. They formulated an objective function to learn a Mahalanobis distance, parameterized by a positive semi-definite matrix M, that maximized the minimal adversarial perturbation on each sample.

Strengths: The paper has dealt with the problem of distance metric learning which is a very common and fundamental problem. The idea is simple and interesting. The paper has demonstrated reasonable improvement over several traditional metric learning methods. The theoretical setting upon which the authors build up their approach is well established.

Weaknesses: The experiment is the major problem in the paper. The authors compared the proposed ARML method with the following baselines: Euclidean, NCA, LMNN, ITML and LFDA. The compared methods are relatively old. The authors used six public datasets on which metric learning methods perform favorably in terms of clean errors, including four small or medium-sized datasets: Splice, Pendigits, Satimage and USPS, and two image datasets MNIST and Fashion-MNIST, which are wildly used for robust verification for neural networks. The first four datasets are very simple and easy to classify. Thus, the practicability of the proposed method is hard to evaluate. Although the observations are interesting, they are rather limited.

Correctness: The solution is technically sound. The claims made by the authors are somewhat supported by their findings.

Clarity: Paper is understandable and relatively well-written.

Relation to Prior Work: The authors discussed how this work differs from previous contributions to some extent.

Reproducibility: Yes

Additional Feedback:


Review 2

Summary and Contributions: The authors present an approach for metric learning that is robust to adversarial perturbations when used in conjunction with K-NN classifier. (Adversarial Robust Metric Learning = ARML). They present an algorithm to learn a Mahalanobis distance (i.e. matrix) M that is robust to additive perturbations to the input. Authors formulate a learning problem to estimate M that approximately maximizes minimal adversarial perturbation. Instead of looking at K neighbors, they consider a triplet loss. For a test point (x), its nearest point of the same class (xp) and its nearest point of a different class (xn), optimize M that decreases the M -distance to the same class point (xp) and increases the distance to other class (xn) normalized by the M -distance between xp and xn. Training is done with SGD over samples of such triplets. AMRL is compared and outperforms a few other approaches. Performance is measured in terms of 1-NN error.

Strengths: Its an interesting paper and a clean and principled formulation of the approximate objective to learn M under adversarial perturbations. Empirical results also show improvement over other algorithms. Since K-NN classifiers are widely used, this is an important area to study and provide mechanism to improve robustness.

Weaknesses: There is no discussion about compuational cost of proposed approach. It would be great to see a comparision table of ARML vs other methods in average training time. Additionally, there is no information about ARML’s effect on classification accuracy. t would be great if there is explaination about their approach vs the existing ones. Some of the missing papers are; “Adversarial Metric Learning” by Chen et al. (2018), “ADVKNN: Adversarial Attacks on K-NN Classifiers with Approximate Gradients” by Li et al. (2019), and “Towards Certified Robustness of Metric Learning” by Yang et al. (2020).

Correctness: Theory and empirical methodology looks correct

Clarity: yes

Relation to Prior Work: Authors discuss related work

Reproducibility: Yes

Additional Feedback: I reviews author's feedback. I appreciate additional comparison to neural network approaches. There is a debate whether deep metric learning is a fair comparison so I am keeping my reviews unchanged.


Review 3

Summary and Contributions: This paper proposes a metric learning method for robust KNN inference against adversarial examples. A minimal norm of adversarial example for metric learning is derived. Neighbor samples are sampled to optimize the minimal norm. The experimental results demonstrate the effectiveness of the proposed method. The results show that the proposed method does not decrease accuracy in the clean setting.

Strengths: 1. A minimal norm of adversarial example for metric learning is derived. And a corresponding optimization scheme is proposed. 2. The results show that the proposed method does not decrease accuracy in the clean setting.

Weaknesses: 1. No metric learning method against adversarial examples is discussed or compared in the experiment. For example: Chen, Shuo, et al. "Adversarial metric learning." Proceedings of the 27th International Joint Conference on Artificial Intelligence. 2018. Mao, Chengzhi, et al. "Metric learning for adversarial robustness." Advances in Neural Information Processing Systems. 2019. Duan, Yueqi, et al. "Deep adversarial metric learning." Proceedings of the IEEE Conference on Computer Vision and Pattern Recognition. 2018. Zheng, Wenzhao, et al. "Hardness-aware deep metric learning." Proceedings of the IEEE Conference on Computer Vision and Pattern Recognition. 2019. 2. The loss function in Eq. (12) is not clear. 3. Sampling instances from the neighborhood of a sample is a important efficiency-related issue, which is not clarified.

Correctness: Yes

Clarity: Yes

Relation to Prior Work: 1. No metric learning method against adversarial examples is discussed or compared in the experiment. For example: Chen, Shuo, et al. "Adversarial metric learning." Proceedings of the 27th International Joint Conference on Artificial Intelligence. 2018. Mao, Chengzhi, et al. "Metric learning for adversarial robustness." Advances in Neural Information Processing Systems. 2019. Duan, Yueqi, et al. "Deep adversarial metric learning." Proceedings of the IEEE Conference on Computer Vision and Pattern Recognition. 2018. Zheng, Wenzhao, et al. "Hardness-aware deep metric learning." Proceedings of the IEEE Conference on Computer Vision and Pattern Recognition. 2019.

Reproducibility: No

Additional Feedback: I am satisified with the rebuttal of the authors.


Review 4

Summary and Contributions: The paper presents a mahalanobis learning algorithm that is certifiable robust to adversarial attacks. The algorithm learns a Mahalabobis matrix which maximizes the minimal adversarial attack on each example. The method is compared against standard learning algorithms on a series of datasets and show that indeed the proposed algorithm has a good robustness to attacks, exhibiting the lowest values of robust error, and often has also the lowest error. To learn the Mahalanobis matrix it defines an objective it establishes a lower bound for minimal adversarial perturbation of some training instance that is parametrized by the Mahalanobis matrix. The bound is based on the minimal perturbation that given an instance and a negative and a positive instance will change the nearest neighbor relation. Using the triplet result one would need to go over all combinations of negative and positive instances for the given instance to compute the value of the bound, resulting in a quadratic complexity. In the actual algorithm this quadratic complexity is reduced by sampling only the nearest neighbors of any given instance, which adds a second level of approximation. For the case of one nearest neighbor they propose a variant of the algorithm that instead of relying on a bound computes the exact minimal adversarial perturbation for any given instance, which involves solving a series of quadratic programming problems in order to get the minimum over them.

Strengths: The paper presents what according to the authors is the first certifiable robust metric learning algorithm. The algorithm that it proposes is grounded on a theoretical established lower bound of the minimum adversarial perturbation and is shown to significantly improve the robustness of metric learning, at least when compared to non-robust standard metric learning baseline.

Weaknesses: Since there are no robust-metrix learning algorithms the algorithm proposed here is compared against standard metric learning algorithms, such as LMNN and ITML. It would have been appropriate to see how the proposed algorithm fairs with respect to robust deep learning algorithms.

Correctness: As I stated before, in the apparent absence of robust metric learning algorithms, it would have been useful to provide results from other families of robust algorithms such as the ones the authors themselves cite, e.g. [43, 11, 48].

Clarity: The paper in general is well written. There were a few points that were not clear to me. I cannot parse equation 6. If I understand well all x_test instances for which |J|=k and |I|=k-1 are missaclassified. The constrain in equation 6 looks for each such instance for the min \delta_{ij}, with j \in i and i in {all same class instances to x_test} - I, that would essentially leave the I and J sets unchanged; in other words we do not want any same class instance of x_test to become a nearest neighbor and kick out a different class nearest neighbor. Shouldn't we be looking instead for all correctly classified instances, i.e. |I|=k, and |J|=k-s, and then among these the minimum perturbation that would reduce the cardinality of I? On the other hand equation 13 which seems to provide the same result for the 1-nn case makes sense. It simply looks for the minimum perturbation over all instances with different class than x that will make one of them the closest instance to x, and thus result in a wrong classification? In addition in equation 7 what do we know for x? should it be that originally it is closer to x+? which would make sense if we would want to find the minimum perturbation that makes it closest to the x-. In the experiments how is the \epsilon value determined in tables 1 and 2? I guess the \epsilon value should somehow depend on the average norm of the training instances?

Relation to Prior Work: yes, the discussion with respect to the previous work is satisfactory, as far as I can judge.

Reproducibility: Yes

Additional Feedback: The response of the authors was satisfactory, and in particular the addition of baselines and the addition of a discussion on related work. I will keep my rating.

[Author Response · NeurIPS 2020]

We appreciate all the reviewers for the constructive comments, and our responses are as below:

**R1.1: concerns on compared methods and datasets.** We fully understand the concern about our baselines since
we are the first to improve certified robustness of metric learning. Therefore, as Reviewer 4 suggested, we add
experiments comparing with neural networks certification methods, including ordinary neural networks certified by
CROWN [48] and randomized-smoothing neural networks [11]. The results are shown in Figure i. Our method
outperforms randomized-smoothing networks for a wide range of $\ell_2$ perturbations, and we will add these baselines to
the paper. As mentioned in the review, these two datasets are often used for evaluating robustness of neural networks, so
surpassing a very recently proposed defensive network (RandSmooth) suggests the potential high impact of this work.
Regarding the datasets, such UCI datasets are actually commonly used for performance evaluation in previous metric
learning papers in various settings (Perrot and Habrard, NeurIPS'15; Zadeh *et al.*, ICML'16; Chen *et al.*, IJCAI'18).

**R2.1: computational cost.** In general, computational
cost is not an issue for ARML. The average runtime
(of 5 trials) of LMNN, ITML, NCA and ARML, are
66.4s, 95.2s, 480.9s and 146.1s respectively, on USPS
for 100 iterations. To make the comparison fair, all of
the methods are run on CPU (Xeon(R) E5-2620 v4 @
2.10GHz). In fact, ARML is highly parallelizable and
our implementation also supports GPU: when running
on GPU (one Nvidia TITAN Xp), the average runtime of
ARML is only 10.6s. We will add detailed discussions.

(a) MNIST.    (b) Fashion-MNIST.

Figure i: Certified robust errors of Mahalanobis 1-NN
compared with ordinary neural networks and randomized-
smoothing neural networks (RandSmooth).

**R2.2: classification error.** We did report the classifi-
cation error. The classification error is equivalent to the
empirical robust error at radius 0 in both Table 1 and Table 2. (Note that the certified robust error in Table 1 is equivalent
to the empirical robust error.) One of the major advantages of ARML over existing metric learning methods is that it
could improve the classification error and robust error simultaneously. We will highlight them in these two tables.

**R2.3 & R3.1: missing papers and existing metric learning methods against adversarial examples.** Thanks for
mentioning these related work. Both Chen *et al.* (2018) and Duan *et al.* (2018) introduced the adversarial framework
as a mining strategy aiming to improve classification accuracy of metric learning, and Zheng *et al.* (2019) made
an improvement upon them; Li *et al.* (2019) proposed an attack method for $K$-NN via a differentiable substitute;
Mao *et al.* (2019) used a metric learning loss as a regularization to improve *empirical* robustness of deep learning
models; A concurrent paper (Yang *et al.*, 2020), made public after the NeurIPS deadline, proposed a similar training
method to ours, but still lacks formal robustness verification like our Theorem 1 and Algorithm 2, and therefore did
not *certify* robustness formally. In contrast, as we underscore in our paper, we propose the first *certified* defense based
on metric learning. We will add a paragraph in the related work section to talk further about adversarial robustness of
metric learning with additional references.

**R3.2: loss function in Eq. (12).** The loss functions in Eq. (11) and Eq. (12) are the same to the one in Eq. (5): a
monotonically non-increasing function. In our implementation, we simply use the "negative" loss, i.e., $\ell(\epsilon) = -\epsilon$. We
will make it clear in the paper.

**R3.3: efficiency issue concerning sampling instances.** Sampling from neighborhood instead of computing $k$th max
and $k$th min is indeed a crucial technique of ARML to help improve training efficiency. We will clarify it.

**R4.1: compare with robust deep learning.** We have compared with robust neural networks (see R1.1).

**R4.2: Eq. (6) and Eq. (13).** It is noteworthy that Eq. (6) involves two layers of optimization, and it is correct that
$\mathbb{I}$ and $\mathbb{J}$ are fixed for the inner optimization. The constraints assure that *at most* $k - 1 = (K + 1)/2 - 1$ "positive"
training instances, i.e., instances in $\mathbb{I}$, are in the $K$-size neighborhood of $\boldsymbol{x}_{\text{test}} + \boldsymbol{\delta}_{(\mathbb{I},\mathbb{J})}$, by means of *not* constraining
$(\boldsymbol{x}_{\text{test}} + \boldsymbol{\delta}_{(\mathbb{I},\mathbb{J})})$'s distances from these "positive" training instances; it is a *necessary* condition for a successful attack.
Interestingly and not surprisingly, for Mahalanobis 1-NN — the case where we have $K = 1$ and $k = (K + 1)/2 = 1$,
and hence $\mathbb{I}$ is an empty set, and $\mathbb{J}$ has only one element — Eq. (6) is exactly reduced to Eq. (13).

**R4.3: selection of $\boldsymbol{x}^+$ in Eq. (7) (or $\boldsymbol{x}_i$ in Eq. (10)).** It is an insightful question. In theory, enumerating every
$\{i : y_i = y_{\text{test}}\}$ in Eq. (10) will derive the *tightest* lower bound of the minimal adversarial perturbation. Nevertheless, in
practice, only selecting a subset of $\{i : y_i = y_{\text{test}}\}$ of which $\boldsymbol{x}_i$ is "close" to $\boldsymbol{x}_{\text{test}}$ usually suffices to derive a satisfying
lower bound. In the robustness evaluation phase, to derive certification bounds as tight as possible, we did not employ
this strategy. We will add more discussions for Theorem 1.

**R4.4: how the radius is determined in experiments.** The radius is only used to show the experimental results, and is
*not* a hyperparameter. In fact, we could also plot the certified robust error curve across "all" radii as in Figure i.

[Meta-Review · NeurIPS 2020]

This paper proposes a metric learning method for robust KNN inference against adversarial examples. Advantages / Main pain points: - First certifiable robust metric learning but lacks comparison to robust metric learning - Authors added in the rebuttal comparison of radius KNN to deep networks and showed good results on mnist and fashion mnist Inconvenient: - Lack of comparison to previous methods doing robust metric learning This paper received mixed initial scores, that sparked a fruitful discussion phase. A consensus emerged between reviewers and AC that the contributions outweigh the execution flaws, and therefore we recommend this work for acceptance. We encourage the authors to add all relevant previous works pointed by reviewers and to add also baselines of robust metric learning.